# Comparative Proteomics and Interactome Analysis of the SARS-CoV-2 Nucleocapsid Protein in Human and Bat Cell Lines

**DOI:** 10.3390/v16071117

**Published:** 2024-07-11

**Authors:** Stuart D. Armstrong, Covadonga Alonso, Isabel Garcia-Dorival

**Affiliations:** 1Department of Infection Biology and Microbiomes, University of Liverpool, Liverpool L3 5RF, UK; sarmstro@liverpool.ac.uk; 2Department Biotecnología, Instituto Nacional de Investigación y Tecnología Agraria y Alimentaria (INIA-CSIC), Carretera de la Coruña km 7.5, 28040 Madrid, Spain; calonso@inia.csic.es

**Keywords:** SARS-CoV-2, nucleocapsid protein, interactome analysis, bat cells, human cells, host–virus interactions

## Abstract

The severe acute respiratory syndrome coronavirus 2 (SARS-CoV-2) is the causative agent of COVID-19 and responsible for the global coronavirus pandemic which started in 2019. Despite exhaustive efforts to trace its origins, including potential links with pangolins and bats, the precise origins of the virus remain unclear. Bats have been recognized as natural hosts for various coronaviruses, including the Middle East respiratory coronavirus (MERS-CoV) and the SARS-CoV. This study presents a comparative analysis of the SARS-CoV-2 nucleocapsid protein (N) interactome in human and bat cell lines. We identified approximately 168 cellular proteins as interacting partners of SARS-CoV-2 N in human cells and 196 cellular proteins as interacting partners with this protein in bat cells. The results highlight pathways and events that are both common and unique to either bat or human cells. Understanding these interactions is crucial to comprehend the reasons behind the remarkable resilience of bats to viral infections. This study provides a foundation for a deeper understanding of host–virus interactions in different reservoirs.

## 1. Introduction

The global health crisis caused by the severe acute respiratory syndrome coronavirus 2 (SARS-CoV-2) has had a profound impact on global health, with over 613 million confirmed cases worldwide. The origin of SARS-CoV-2 has been the subject of intense investigation and debate [1]. Bats, as natural reservoirs for various coronaviruses, have been proposed as potential sources for SARS-CoV-2 [2,3]. Understanding the role of bats in SARS-CoV-2 transmission is crucial, given their unique immune adaptations and their historical association with several coronaviruses [4].

Bats are one of a diverse group of mammals recognized for their ability to host a wide range of viruses, often without displaying clinical symptoms. The association between bats and coronaviruses is not new; notable examples include Middle East respiratory syndrome coronavirus (MERS-CoV) and severe acute respiratory syndrome coronavirus (SARS-CoV). Although the zoonotic transmission of viruses from bats to humans is rare, it has gained attention due to its potential to trigger global health crises [5]. In this context, our study focuses on a comparative analysis of the SARS-CoV-2 nucleocapsid (N) protein interactome in human and bat cell lines.

The N protein, or nucleocapsid protein, is one of the most abundant viral proteins. It plays a pivotal role in the viral life cycle and has several important functions, such as its role as a structural component and packaging of the viral genome [6]. It is also known for its RNA binding properties, but still has several unknown functions that required further elucidation [7,8,9]. Due to the importance of this protein in the viral cycle as well as the fact that the N protein is one of the most abundant viral proteins, it has been considered particularly attractive as an antiviral target [10]. Furthermore, several studies indicate that mutations in this viral protein can increase virulence and infectivity [11,12,13]. All these studies highlight the importance of the study of the role of the N protein as well as its interactome in the viral life cycle.

Virus replication relies on the complex protein–protein interaction (PPI) network formed by specific viral–host interactions. Although some proteomic studies have mapped the PPI network between some selected SARS-CoV-2 proteins and human proteins [14,15,16,17], only a few studies have been conducted in bat cells. In this work, a comparative analysis of the N protein interactome in both bat and human cell lines is conducted. This may contribute to a better understanding of SARS-CoV-2 biology and provide insights into viral adaptation mechanisms. It is important to note that the precise mechanisms underlying the transmission of certain coronaviruses from bats to humans and other animals are still unknown [5,18]. It is therefore crucial to study and identify the distinctive and shared cellular factors among these species that may facilitate infection [19].

The objective of this investigation is to contribute to a broader understanding of SARS-CoV-2 biology and the intricate dynamics of host–virus interactions. This will provide insights into potential adaptations of the virus. By examining the similarities and differences in the N protein functions between human and bat cells, we aim to elucidate key aspects of the viral life cycle and host response. This will provide insights for future research in understanding host–virus interactions, drug development and therapeutics, as well as public health preparedness.

A total of 168 and 196 cellular proteins were identified as having a high probability of interacting with SARS-CoV-2 N in human and bat cells, respectively. Our observations suggest that the N protein exhibits comparable biological properties in both human and bat cells. Bioinformatic analysis revealed that some of these cellular interactors are involved in the ribosome process, ribonucleoprotein complex biogenesis process and mRNA metabolic process, among others. Nevertheless, our study also identifies some differential host factors, which suggests potential differences in N protein functions between the two species. The objective of this comparative interactome analysis of the N protein in human and bat cells is to elucidate potential differences and provide insights into the host–virus interactions associated with SARS-CoV-2.

## 2. Materials and Methods

### 2.1. Cell Culture

Human embryonic kidney 293T cells (ECACC 12022001) were cultured in complete DMEM (Dulbecco’s modified Eagle’s medium) at 37 °C and 5% CO_2_ atmosphere, with the addition of 100 IU/mL penicillin, 100 µg/mL streptomycin, 1× GlutaMAX (Thermo Scientific, Altrincham, UK) and 10% heat-inactivated fetal bovine serum (FBS). The cell line RoNi/7, derived from kidney cells of the bat *Rousettus aegyptiacus,* was cultured in DMEM with 4.5 g/L glucose (PAA) and supplemented with 10% FBS, 1% penicillin/streptomycin 100 × concentrate (penicillin 10,000 U/mL, streptomycin 10 mg/mL), 1% GlutaMAX, 1% sodium pyruvate 100 mM (Thermo Scientific, Altrincham, UK) and 1% MEM non-essential amino acids (NEAA) 100 × concentrate (Thermo Fisher) and grown at 37 °C and 5% CO_2_. The bat cell line was kindly provided by Prof. Christian Drosten (EVAg, Marseille, France).

### 2.2. Design and Construction of Plasmid That Expresses SARS-CoV-2 N

To design the EGFP-SARS-CoV-2 N plasmid, a codon-optimized cDNA sequence of the ORF of SARS-CoV-2 N (NCBI accession number: NC_045512) was cloned into the pEGFP-C1 (by GeneArt-Thermo Fisher Scientific). After cloning, the plasmid sequence was validated through sequencing by Gene Art (Thermo Fisher Scientific; Regensburg, Germany).

### 2.3. Transfection and Expression of SARS-CoV-2 N and EGFP in Bat and Human Cells

Some of the methodology described in this paper was previously described by García-Dorival et al. 2014 and García-Dorival et al. 2016 [20,21]. To transfect HEK 293T cells, 1 × 10^7^ cells were seeded 24 h prior transfection. The calcium phosphate transfection method was performed with 25.6 μg of plasmid DNA being used for each plasmid. For RoNi/7 cells, 1 × 10^7^ cells were seeded 24 h prior to transfection, after which transfection was performed using FuGene (Promega, Madison, Wisconsin, United States) according to the manufacturer’s instructions. Both cell lines were harvested 24 h post transfection, lysed, and coimmunoprecipitated using a GFP-Trap® (Chromotek, Planegg, Germany).

### 2.4. Immunoprecipitations

EGFP-N fusion protein and EGFP immunoprecipitations (IPs) were performed using GFP-Trap^®^_A (Chromotek) according to the manufacturer’s instructions. For immunoprecipitations, the cell pellet was resuspended in 200 μL of lysis buffer (10 mM Tris/Cl pH 7.5; 150 mM NaCl; 0.5 mM EDTA; 0.5%NP40) and then incubated on ice for 30 min. Subsequently, the lysate was then clarified by centrifugation at 14,000× *g* and diluted 5-fold with dilution buffer (10 mM Tris/Cl pH 7.5; 150 mM NaCl; 0.5 mM EDTA). The GFP-Trap agarose beads were equilibrated with ice-cold dilution buffer and then incubated with diluted cell lysate overnight at 4 °C on a rotator, followed by centrifugation at 2500× *g* for 2 min. The bead pellet was washed twice with wash buffer (10 mM Tris/Cl pH 7.5; 150 mM NaCl; 0.5 mM EDTA). After the removal of the wash buffer, the beads were either resuspended in 100 μL of sample buffer, Laemmli 2× concentrate (Sigma Aldrich, Gillingham, UK) and boiled at 95 °C for 10 min to elute the bound proteins or resuspended in 50 mM Tris HCl for analysis by label-free mass spectrometry. All buffers used for immunoprecipitations were all supplemented with Halt^TM^ Protease Inhibitor Cocktail EDTA-Free (Thermo Scientific, Altrincham, UK).

### 2.5. Peptide Preparation and Protein Identification by LC-MS/MS

GFP-Trap beads were washed three times with 50 mM Tris HCl, pH 7.5, before incubation with digest buffer (50 mM Tris HCl, pH 7.5, 2 M urea, 5 µg/µL Trypsin (Sigma), 1 mM DTT) at 30 °C and 400 rpm for 30 min. The beads were then pelleted by centrifugation at 2.5× *g* for 2 min at 4 °C. The supernatant was then transferred to a fresh low-binding tube (Eppendorf, Stevenage, UK). The beads were then resuspended in elution buffer (50 mM Tris HCl, pH 7.5, 2 M urea, 5 mM iodoacetamide) and centrifuged at 2.5× *g* for 2 min at 4 °C. The supernatant was then combined with the digest buffer. This elution was repeated a further two times. Digestion was continued by incubation at 32 °C and 400 rpm overnight. The reaction was stopped by the addition of 1 µL of trifluoracetic acid (TFA). Samples were desalted using Pierce™ C18 Spin Tips, according to the manufacturer’s instructions. Eluted peptides were reduced to dryness using a centrifugal vacuum concentrator (Eppendorf) and re-suspended in 3% (vol/vol) methanol and 0.1% (vol/vol) TFA for analysis by MS.

Liquid chromatography–mass spectrometry (LC-MS/MS) analysis was performed as described by Alruwaili et al., 2023 [22]. Peptides were analyzed by online nanoflow LC using the Ultimate 3000 nanosystem (Thermo Scientific, Altrincham, UK) equipped with an Easy-Spray PepMap analytical column (RSLC 50 cm × 75 µm inner diameter, C18, 2 µm, 100 Å). The column was operated at a constant temperature of 35 °C. The LC system was coupled with a Q-Exactive mass spectrometer (Thermo Scientific, Altrincham, UK). Chromatography was performed with a buffer system consisting of 0.1% formic acid (buffer A) and 80% acetonitrile in 0.1% formic acid (buffer B). The peptides were separated by a linear gradient of 3.8–50% buffer B over 30 min at a flow rate of 300 nL/min. The Q-Exactive was used in data-dependent mode, with survey scans acquired at a resolution of 70,000 at *m*/*z* 200. The scan range was 300–2000 *m*/*z*. Up to the top 10 most abundant precursor ions with charge states from +2 to +5 from the survey scan were selected with an isolation window of 2.0 Th and fragmented by higher energy collisional dissociation with an NCE of 30. The maximum ion injection times for the survey scan and the MS/MS scans were 250 and 100 ms, respectively, and the ion target value was set to 10^6^ for the survey scans and 10^5^ for the MS/MS scans. MS/MS events were acquired at a resolution of 35,000. Repetitive sequencing of peptides was minimized through dynamic exclusion of the sequenced peptides for 20 s.

MS spectra data were analyzed by label-free quantification using MaxQuant software (version 2.1.3, Cox et al. 2014) [23] using default settings. Data were searched against a human protein database (Uniprot, UP000005640_9606, Nov 2023) or an *R. aegyptiacus* database (Uniprot, UP000593571_9407_Nov23) together with the SARS-CoV-2 N bait protein (Uniprot accession: P0DTC9). The search included variable modifications of methionine oxidation and N-terminal acetylation and a fixed modification of carbamidomethyl cysteine. Enzyme specificity was set to trypsin, and a maximum of two mis-cleavages were permitted. The false discovery rate (FDR) was set to 0.01 for peptide and protein identifications.

### 2.6. Label-Free Mass Spectrometry and Bioinformatics Analysis

Mass spectrometry analysis was performed in triplicate for EGFP-N, and EGFP in human and bat cells. Once the label-free mass spectrometry results had been processed, the Perseus software was used for the statistical analysis of the data. This approached enabled the differentiation of background proteins (the cellular proteins interacting with EGFP or the matrix alone) from the interacting proteins (those cellular proteins interacting with either SARS-CoV-2 N). LFQ intensity values were analyzed using Perseus v1.6.15.0 software (Max Planck Institute, Munich, Germany). A t-test sample analysis was performed with a *p*-value of 0.05, resulting in the generation of a volcano plot and tables, which identified those proteins with the highest probability of interacting with SARS-CoV-2 N.

### 2.7. Western Blot Analysis

Western blot analysis was conducted to confirm the expression of EGFP and EGFP-N proteins. SDS-PAGE was performed, followed by Western blot (WB) analysis, as previously described in [24]. Mini-PROTEAN TGX gels (Bio-Rad, Hercules, CA, USA) were used for SDS-PAGE, and then the gels were transferred to PVDF membranes using the Trans-Blot Turbo Transfect Pack (Bio-Rad, Hercules, California, United States) and the Trans-Blot Turbo system (Bio-Rad). Subsequently, the membranes were incubated in 10% skimmed milk powder in TBS-0.1% Tween (TBS-T) buffer (50 mM Tris-HCl (pH8.3), 150 mM NaCl and 0.5% (*v*/*v*) Tween-20) for one hour at room temperature. The primary antibody was diluted in 5% skimmed milk powder dissolved in TBST at a dilution of 1:1000 and then incubated at 4 °C overnight. After three washes, the blots were incubated with the appropriate HRP secondary antibody, diluted in 5% skimmed milk powder dissolved in TBST at a concentration of 1:5000 for one hour at room temperature. Subsequently, the blots then were developed using an enhanced chemiluminescence reagent (Bio-Rad, Hercules, California, United States) and detected using the ChemiDoc™ XRS Gel Imaging System with the Image Lab™ software (Bio-Rad, Hercules, CA, USA).

### 2.8. Bioinformatic Analysis of SASR-CoV-2 Interacting

The interactions of SARS-CoV-2 N-interacting proteins were visualized using Metascape software (http://metascape.org; v3.5.20240101) [25]. In order to elucidate the functional roles of the identified protein complexes, several analyses were performed. These included pathway and process enrichment analysis, protein–protein interaction enrichment analysis and quality and control association analysis.

#### 2.8.1. Functional and Enrichment Analysis

For the functional and enrichment analysis, Metascape was used with the default settings. As previously in Zhou et al., 2019 [25]. Metascape employs the hypergeometric test and Benjamin–Hochberg *p*-value correction algorithm to identify all ontology terms that contain a statistically significant number of genes in common with an input list in comparison to what would be expected by chance. The pathway enrichment analysis in this software incorporates a number of databases, including Gene Ontology, KEGG, Reactome, MSigDB, and others. The enriched terms are then automatically clustered into a non-redundant group using a logic similar to that employed by DAVID. The pairwise similarities between enriched terms are calculated based on a Kappa test score. The resulting similarity matrix is hierarchically clustered, and a 0.3 similarity threshold is applied to segment the tree into distinct clusters. The most significant term within each cluster (with the lowest *p*-value) is selected to represent the cluster in bar graphs and heatmaps. Furthermore, the analysis incorporates other prevalent enrichment metrics in addition to *p*-values.

#### 2.8.2. Comparative Interactome Analysis

For the interactome analysis of both datasets, Metascape software was also employed. By default, Metascape utilizes BioGrid’s physical protein–protein interaction data as the primary source for interactome analysis. Furthermore, it incorporates more recent human interactome datasets, such as InWeb_IM and OmniPath, to enhance interactome coverage. Upon inputting a list of proteins, Metascape automatically generates a protein interaction network from these candidates. The MCODE algorithm is applied iteratively to each connected network component, with performance modifications, in order to identify densely connected complexes. Subsequently, functional enrichment analysis is conducted on each complex, with the top three enriched terms used to annotate its biological role.

#### 2.8.3. Comparative Analysis

To provide biological context to our SARS-CoV-2 N interactome dataset for both species, all previously mentioned analyses were applied. Additionally, to compare our dataset with other studies, Coronascape, a tool within Metascape software, was used [25]. Coronascape facilitated a comparative analysis of our interactome with those of other SARS-CoV-2 interactome studies.

Orthologous genes were found using Proteinortho [26], using default settings.

## 3. Results

### 3.1. Expression of SARS-CoV-2 N in 293T Cells and RoNi/7 Cells

To investigate the potential cellular interacting partners of SARS-CoV-2-N, a high-affinity EGFP immunoprecipitation system (GFP-Trap) (Chromotek, Planegg, Germany) (co-IP) coupled with a label-free mass spectrometry approach was used (Figure 1A). The SARS-CoV-2 N protein was expressed with an EGFP protein tag and then immunoprecipitated using the GFP-Trap system from Chromotek. This system has been demonstrated to improve the sensitivity and enable the differentiation between specific and non-specific interactions with the target protein [16]. The gene encoding for a codon-optimized SARS-CoV-2-N was cloned at the 3′ terminal of EGFP (EGFP-N), creating a contiguous open reading frame and expressing the fusion protein.

EGFP-N and the control EGFP were overexpressed in bat/human cells and then extracted from lysed cells for immunoprecipitation. The eluate fraction obtained from the immunoprecipitation was then analyzed by label-free mass spectrometry (MS) in order to identify potential interacting partners of SARS-CoV-2-N. To reduce the false-positive rate, EGFP alone was also expressed in both cell lines as a control.

In order to study the interactome in human cells, 293T cells were selected due to their high transfection efficiency. To study the interactome of bat cells, we selected RoNi/7 cells from fruit bats *Rossetus aegyptiacus* due to the availability of the annotation database as well as its susceptibility to SARS-CoV-2 infection, as demonstrated in experimental studies in vivo [27]. Protein expression in both cell lines was confirmed using Western blot and immunofluorescence.

The potential interaction partners for SARS-CoV-2-N were immunoprecipitated using the EGFP-Trap system. Subsequently, a Western blot analysis was then conducted on both the input and elution (or bound) samples. An anti-EGFP antibody was used to detect EGFP- SARS-CoV-2-N (170 kDa) (Figure 1B). Label-free MS and quantitative proteomics were then employed to differentiate between the EGFP-N interactomes and the control (EGFP alone), leading to the identification of possible interacting partners of SARS-CoV-2-N.

### 3.2. Identification of the Interacting Partners of SARS-CoV-2 N in Humans and Bat Cells

As in previous publications from our group [19,21,28], the label-free mass spectrometry results were processed and analyzed using the Perseus software (MaxQuant; version v2.1.0.0). The software was used to perform the statistical analysis and to differentiate background proteins (cellular proteins that interacted with EGFP alone) from interacting proteins (cellular proteins that interacted with SARS-CoV-2-N in either human or bat cells).

The LFQ intensity values were analyzed using the Perseus software platform, version 2.0.6.0 [29], as previously described [21,28]. To eliminate false positives, a *t*-test analysis was performed with a *p*-value of <0.05, along with an FDR of <0.05. Furthermore, proteins identified with a single peptide were excluded from analysis.

Following the statistical analysis, 196 and 168 proteins were identified as having a high probability (95% confidence) of interacting with N in bat and human cells, respectively (see to Appendix A). Following the statistical analysis, the data were presented in the form of volcano plots (Figure 2). The logarithmic ratio of protein intensities (on the *x*-axis) is plotted against the negative logarithmic *p*-values of the t-test (on the *y*-axis) (Figure 2) in these volcano plots.

Each dot on the right upper quadrant of the volcano plot represented a protein with a high probability of interacting with the SARS-CoV-2-N. Significant proteins are defined as those with a *p*-value < 0.05 and a fold change of around 2 or higher.

A total of 98 cellular proteins were identified as common interactors between the two species, as detailed in Appendix A. Examples of common interactors include ribosomal proteins, interleukin enhancer binding factor 3 (ILF3), DEAD-box helicase 5 (DDX5) and DExD-box helicase 21 (DDX21; also known as DEAD-box helicase 21), among others. A number of these cellular proteins have already been described as important for SARS-CoV-2, which gives confidence to our analysis and supports our study [30]. Furthermore, 97 unique cellular proteins were identified in the bat cell interactome, while 70 cellular proteins were found to interact exclusively with human cells. An example of this is the ADAR protein, which has been identified exclusively in the N SARS-CoV2 human interactome. This protein has been linked to the induction of RNA editing in SARS-CoV2 [31,32]. Further examples of protein interactors that are unique to the human cell line are heat shock protein 70 (HSP70) and the Karyopherin subunit alpha 2 (KPNA2) protein, which have been reported to be linked to SARS-CoV-2 infection [33,34].

As examples of unique cellular interactomes in bat cells, we identify the co-chaperones DNAJA2 and DNAJA3, as well as the nuclear cap-binding protein subunits 1 and 2, among others. The protein DNAJA2 has previously been reported to interact with SARS-CoV-2 in human cells [35].

### 3.3. Enriched Genes Involved in Different Cellular Pathways for Bats and Humans

Following the identification of the potential N-SARS-CoV-2 interactors, a functional enrichment analysis was performed to further explore the functions of these host cellular proteins. For this analysis, all statistically enriched terms were first identified using cumulative hypergeometric *p*-values, and enrichment factors were calculated and used for filtering according to the parameters set out in Metascape software. Numerous protein components related to diverse pathways were identified using bioinformatic tools.

A functional analysis of the SARS-CoV-2 N interactome revealed both unique and shared enriched domain terms for both the bat and human datasets when analyzed separately (Figure 3). Among the top 20 enriched functional terms (GO and KEGG) across the N SARS-CoV-2 interactome for both species, we observed common pathways related to ribosome and cytoplasmic. We have also observed for both species terms related to ribonucleoprotein complex biogenesis and the regulation of translation, among others (Figure 3A,B). The terms within each cluster for both interactomes can be found in Appendix A.

### 3.4. Comparison of the N Protein Interactome in Bats and Humans

In order to gain further insight into the biological meaning of the N-SARS-CoV-2 interactors in both species, a bioinformatic and functional analysis of both interactomes was performed.

First, we determined the common and unique proteins for both interactomes using the Venny 2.1 program [36] and Metascape in order to identify how many proteins were shared by both interactomes (Figure 4A,B). A total of 98 interacting proteins were identified in both interactomes, with several classes of proteins (Appendix A). The most abundant proteins common to both interactomes were ribosomal proteins, DEAD-box helicases (DDX) and heterogeneous nuclear ribonucleoproteins (Appendix A). Some of these proteins have been described as important for SARS-CoV-2, including DDX21 [30]. Other proteins identified in this study have also been identified by other authors as interacting with N-SARS-CoV-2 such as HNRPU, RPS8, RPS6, LARP-1 and G3BP1, among others [9].

Additionally, enrichment analysis was conducted on both interactomes. The terms with the lowest *p*-values within each cluster were selected as representative terms, and a dendrogram was generated to display them. The heatmap cells with the selected GO top 100 (Figure 4C) are colored by their *p*-values. Gray cells indicate the absence of enrichment for that term in the corresponding gene lists. A complete list of the proteins included in each cluster for the selected GO top 20 can be found in Appendix A.

Among the ontology clusters that were found to be enriched across studies, the most significant ones were those related to mRNA metabolic processes, regulation, and splicing, as well as those related to ribosome and ribonucleoprotein complex biogenesis and protein–RNA complex assembly (Figure 4C). Some of these enriched functional terms have been previously described for SARS-CoV-2 N [10]. Furthermore, we identified a number of selected GO terms related to the immune response and SARS-CoV-2 infection and the innate immune response and antiviral mechanism by IFN-stimulated genes, among others (Figure 4C). While some of these selected GO terms were common to both interactomes, others were only found in either the human or bat interactomes. Some examples of the selected GO found only in the bat interactomes were related to viral translation, VLDLR internalization and degradation, among others (Figure 4C). Furthermore, a subset of representative terms from the full cluster and their conversion into a network layout can be seen in (Appendix A). In addition to this, as a part of the analysis for both interactomes, a comparison with other COVID interactomes datasets was performed using Metascape software as part of our comparison analysis. This facilitated a comparative analysis of our interactome with other SARS-CoV-2 interactome studies (Appendix A).

## 4. Discussion

A number of studies have indicated a potential link between SARS-CoV-2 and other coronaviruses with bats [1,37,38]. Furthermore, the prevailing scientific consensus suggests that SARS-CoV-2 likely emerged from bats, with an intermediary species hypothesized to have facilitated its transmission to humans [2,39]. Research indicates that coronaviruses can infect various bat cell lines. For example, a study involving fruit bats (*Rousettus aegyptiacus*) demonstrated that these bats can harbor SARS-CoV-2 without exhibiting any discernible symptoms, thus suggesting their potential role as a reservoir host [27,40].

To elucidate the differences in infection outcomes between humans and the potential viral reservoir in bats, it is essential to comprehend the manner in which cellular proteins interact with SARS-CoV-2 and how they have evolved in both species. In this study, we conducted a comparative proteomic analysis of the N-SARS-CoV-2 protein interactome. An understanding of the differences in interactomes between both species may prove to be the key to elucidating the barriers to interspecies jumps, as well as the in vivo pathogenic differences. This study represents a preliminary attempt to identify the specific virulence factors for both species. However, further studies are required to confirm and further validate the importance of these findings.

Our findings indicate that while both species share some common interactors, such as several ribosomal and nuclear ribonucleoproteins, they also have unique cellular proteins capable of interacting with N SARS-CoV-2 in either bat or human cells. Additionally, we identified distinct pathways for each species.

Some examples of common cellular interactors include RPL36, a ribosomal protein associated with RNA processing. While this protein has previously been identified as a potential interactor of SARS-CoV-2 in human cells [15], this study marks the first description of its potential interaction with SARS-CoV-2 in bat cells. Furthermore, other proteins related to RNA processing, including DDX21, LARP1, G3BP1, and G3BP2, corroborate findings from previous studies [15,16], reinforcing the reliability of the study’s results. Another notable example of common cellular interactors is the family of DEAD-box (DDX) RNA helicase proteins. Previous research has indicated that this protein family may play a role in the pathogenesis of SARS-CoV-2 infection [41]. Moreover, specific members of this family, such as DDX-5, have been demonstrated to interact with SARS-CoV infection [42,43].

As an example of the specific cellular interactome for human cells, we have previously mentioned the ADAR protein. This enzyme catalyzes the hydrolysis of adenosine to inosine in double-stranded RNA (dsRNA), a process referred to as A-to-I RNA editing [44]. ADAR proteins play a pivotal role in the innate immune response to viral infections, where editing can have a range of pro- or antiviral effects and can contribute to viral evolution [45]. The significance of this protein has also been investigated in the context of SARS-CoV-2 infection [31,32,46,47]. This study highlighted also the unique cellular interactome observed in bat cells, which included the co-chaperones DNAJA2 and DNAJA3. The protein DNAJA2 has previously been reported to interact with SARS-CoV-2 but not with the nucleoprotein [35].

Additionally, PABPC1 was identified as the most significant cellular protein in the N-SARS-CoV-2 bat interactome. PABPC1 binds the poly(A) tail of mRNA, including that of its own transcript, and regulates processes of mRNA metabolism, such as pre-mRNA splicing and mRNA stability [48]. It is well established that this family of proteins represents a common target for viruses, including coronaviruses [49]. It has been reported that the N protein of the coronavirus porcine epidemic diarrhea Virus (PEDV) interacts with PABPC1. This interaction results in the inhibition of PEDV replication [50]. In certain coronaviruses, PABPs can bind to the viral genome poly(A) tail, which may facilitate virus propagation [49]. We also observed that some of the interactors we found, for example, for the human analysis, were also found in previous studies. Examples includes the heterogeneous nuclear ribonucleoprotein U (HNRNPU) and the 40S ribosomal protein S12 (RPS12), among others [14].

Furthermore, it was observed that for the majority of the interacting partners, the cellular proteins that were unique to each construct generally exhibited a lower fold enrichment compared to those that interacted with both constructs. However, we have also identified unique interacting partners with a higher fold change specific for each species. For example, poly(A) binding proteins cytoplasmic 1 (PABPC1s) were found to interact with the bat interactome or several histones that were found for the human interactome. Appendix A section provide a complete list of the potential interacting partners for SARS-CoV-2 N in bats and humans.

To gain a deeper insight into the N-SARS-CoV-2 interactomes for both species, we employed bioinformatic tools. We identified several unique and common pathways for both species, in addition to differences and similarities among the interactomes. Some of these pathways have already been identified as playing a crucial role in viral infections, particularly those related to the immune response. Furthermore, we have also compared both interactomes with previous interactome studies for SARS-CoV-2 using bioinformatic tools. As a result, we identified some common pathways that have been previously identified by other authors [16,51,52]. Some of these studies have been conducted in different cell lines, at different time points, or even using different methods to detect the interactome. This provides reassurance regarding the reliability of our findings as some of the results are comparable. However, we also acknowledge the discrepancies that were identified. To enhance the credibility of our results, multiple replicates were performed in our study. This approach allowed us to perform a more robust statistical analysis, which also gave confidence in the results obtained.

This study also identified typical pathways related to viral infections, including TNF-alpha NF-kappa B signaling and the activation of antiviral mechanisms by IFN-stimulated genes. Furthermore, we found highly significant pathways that are common to both interactomes, including the TRBP-containing complex and the DICER complex. Additionally, several pathways related to RNA processing or regulation were identified as among the most significant interactor proteins for both species. Examples of this category include mRNA metabolic processes, the regulation of mRNA metabolic processes, the regulation of RNA splicing, and ribonucleoprotein complex biogenesis. The regulation of mRNA metabolism is of significant importance for several viruses, including influenza [53] and SARS-CoV-2 [54]. However, the precise function of this process during infection remains to be determined. Furthermore, more specific pathways were identified for each species, including those related to viral translation and interferon-gamma signaling in the N SARS-CoV-2 bat interactome.

This research is the starting point for comparative proteomics analysis between humans and bats. In this study we mainly use bioinformatic tools; however, we do understand further validations should be carried out to confirm the interactors, especially for the bat interactome. The lack of some reagents necessary for working with bat samples presents a challenge in the validation techniques. To gain a deeper understanding of the cross-species transmission barriers and variations in the infectious cycle of the SARS-CoV-2 virus, it is essential to conduct further investigations into the specific functions of certain species-specific proteins.

## Figures and Tables

**Figure 1 viruses-16-01117-f001:**
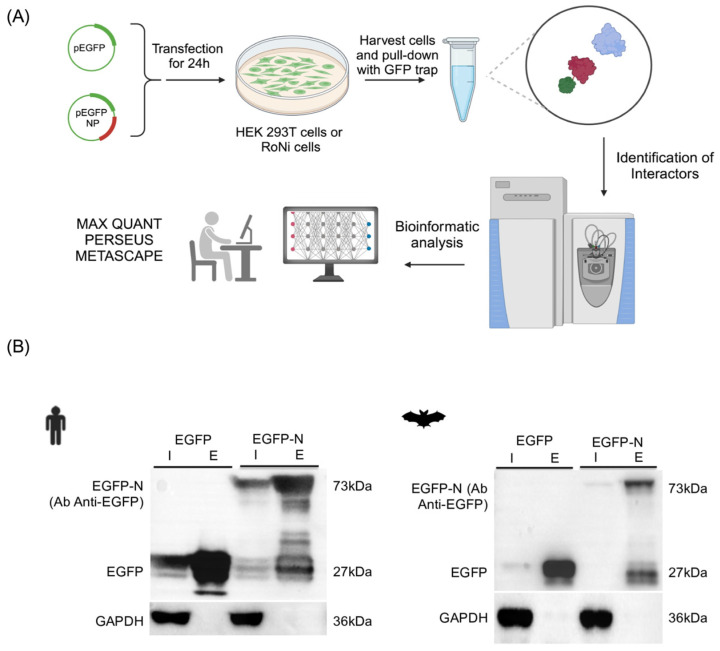
(**A**) Schematic representation of the methodology in this study. HEK 293T and RoNi-7 cells were grown in complete DMEM at 37 °C with 5% CO_2_. At 24 h post transfection, the cells were harvested, lysed, and immunoprecipitated using GFP-Trap (Chromotek). Subsequently, label-free mass spectrometry analysis was conducted on the eluted samples, followed by bioinformatic analysis. (**B**) Expression and co-immunoprecipitation of GFP-tagged SARS-CoV-2 N (GFP-NP) in human HEK293T and bat RoNi7 cell lines. The detection of GFP-tagged SARS-CoV-2 N proteins and control GFP in the immunoprecipitation assay by Western blot (WB) analysis. The letter “I” refers to the input sample and “E” to the elution sample. Cell lysates from transfected cells were subjected to co-immunoprecipitated with agarose beads.

**Figure 2 viruses-16-01117-f002:**
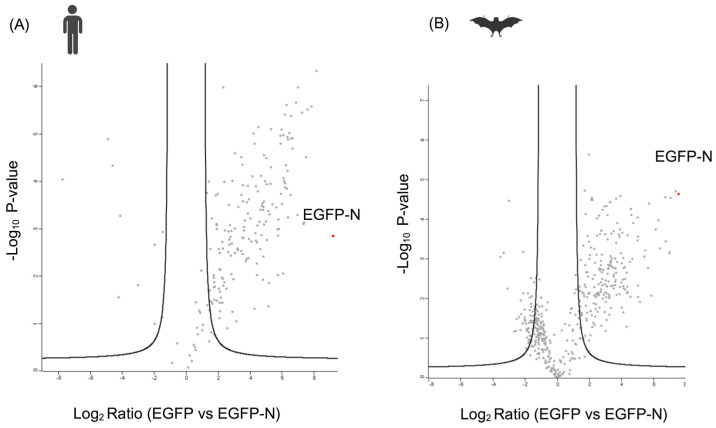
Bioinformatic and statistical analysis of the interactome partners of SARS-CoV-2 N (GFP-NP) protein for both cell lines. Heatmap of the samples (triplicates) used for the analysis for both (**A**) human and (**B**) bat cells. Volcano plots representing mass spectrometry and statistical analysis results for SARS-CoV-2 N protein for (**A**) human and (**B**) bat cells. Analysis by MS included EGFP-tagged SARS-CoV-2 N and EGFP control. The immunoprecipitation and label-free mass spectrometry analyses were performed in triplicate. The logarithmic fold change was plotted against the negative logarithmic *p* values of the *t*-test. In the volcano plots, the dashed curve indicates the region of significant interactions; the upper right quadrant dots represent potential protein-interacting partners. The abundance values of any potential protein interaction partner were compared when co-immunoprecipitated with the EGFP-N fusion protein to their values from the immunoprecipitation with the EGFP control.

**Figure 3 viruses-16-01117-f003:**
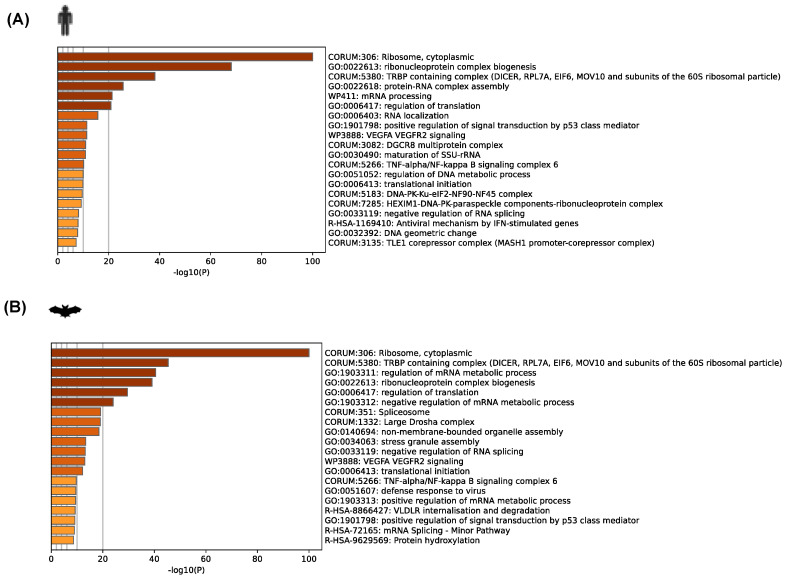
Bioinformatic analysis of the interactome partners of SARS-CoV-2 N (GFP-N) protein for both cell lines corresponding to human (**A**) and bat cells (**B**). Analysis by MS included EGFP-tagged SARS-CoV-2 N and EGFP control. The bar graph displays the enriched terms across the input gene list for SARS-CoV-2 N-interacting partners for human and bat. The bars were colored with intensities according to their *p*-values. The bar graph displays the top 20 enriched terms (including GO/KEGG terms, canonical pathways, hallmark gene sets, etc.) for proteins, colored according to their −log10 (*p*-value).

**Figure 4 viruses-16-01117-f004:**
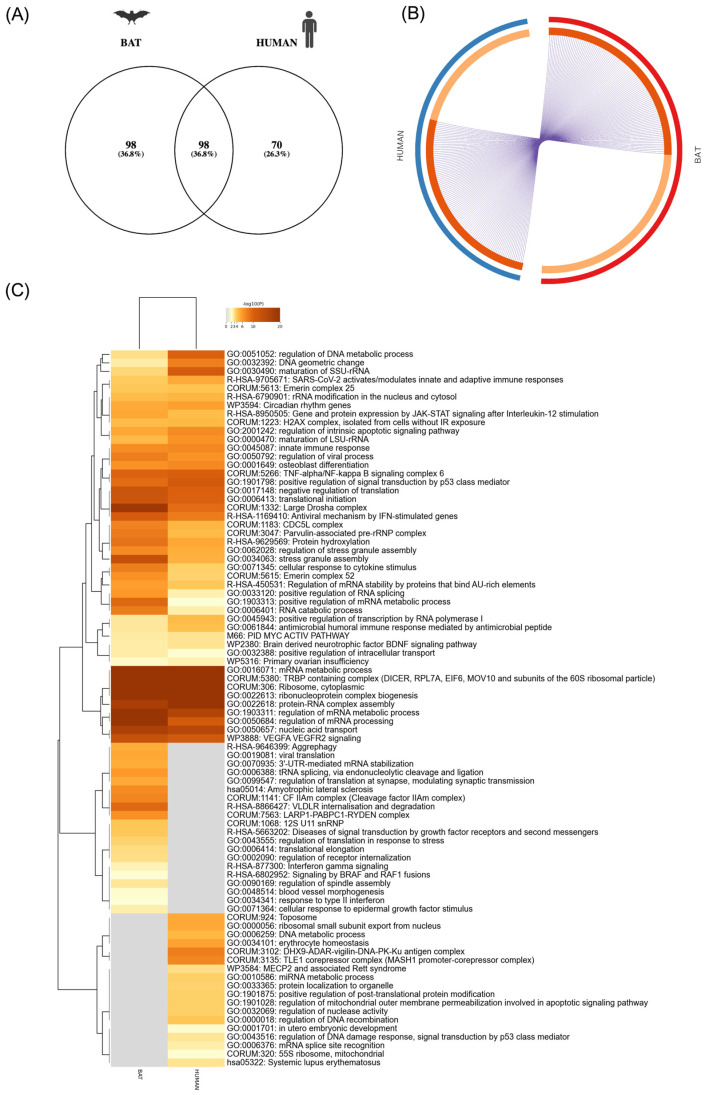
Bioinformatic and functional analysis of the interactome partners of SARS-CoV-2 N (GFP-NP) using Metascape software. Visualizations of meta-analysis results based on the interactomes of SARS-CoV-2 N protein for both cell lines. (**A**) Venn diagram showing common interacting partners among cell lines. (**B**) A Circos plot shows the overlap of genes from the input gene list. Each arc represents one gene list. The dark-orange color represents the genes that are shared by multiple lists, while those unique to a gene list are represented by light orange. Purple lines indicate the presence of a gene in multiple gene lists. (**C**) A heatmap of enriched terms across both input gene lists (SARS-CoV-2 N-interacting partners for human and bat cells), with colors indicating *p*-values. The heatmap cells are colored according to their *p*-values, with gray cells indicating the absence of enrichment for that term in the corresponding gene list.

## Data Availability

All data are contained in the manuscript, figures and Appendix A.

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
