# Peer review of "Comparative Proteomics and Interactome Analysis of the SARS-CoV-2 Nucleocapsid Protein in Human and Bat Cell Lines"

_viruses, 2024, doi:10.3390/v16071117_

Round 1
Reviewer 1 Report
Comments and Suggestions for Authors
Armstrong et al. submit a work entitled ”Comparative proteomics and interactome analysis of the SARS CoV-2 nucleocapsid protein in human and bat cell lines".
The work reports proteomics studies of the HEK293T cellular protein partners of the SARS-CoV-2 N protein and compares the results with the same kind of studies performed on the RoNi/7 kidney cells of a bat species.
The work is original with their comparison of two different species and interesting with identification of proteins unique to a specific genome but we think that the presentation of the results should be improved for several points.
-A more reasoned discussion about the comparison of the results for the human genome with those obtained in previous works. A recent work in the field ant that is reported in the text (Min, Huang et al., 2023, Moll Cell proteomics) underline the rather larger discrepancy of the results for proteomics studies of N SARS-CoV-2 partners between several studies. They note the importance of the cell line, type of method. While the authors mentioned in different parts of the text these previous works, we think that their discussion of the comparison of their data with these latter must be reinforced to enlighten the reader about the relevance and the relativity of the obtained results.
-It is not clear for the reader if the interactions between protein N and the cellular proteins that are reported could arise or not due to the binding of N protein to nucleic acids that are themselves bound by the others RNA binding proteins. Is it possible to reject this possibility with the used protocol?
-we note that there is no validation of the proteomic data using independent pulldown and immunoblot analysis with a set of identified proteins similarly to previous works on the subject. We agree that the work is a short communication and that it is indicated in several places of the text that it is a preliminary study but the authors should argue why these controls have not been performed
Minor points:
-The presentation of the N protein in the Introduction is a little too quick, we think it is important to argue about the importance of the protein and the fact that mutations in protein N have been found to be associated with increased infectivity of viral strains, and we suggest to add references on this topic as possible references concerning this point : Syed, AM, Taha, TY et al., 2021, Science ; Wu H, Xing, N. et al. 2021, Cell Host & Microbe; Johnson, BA, Zhou, Y et al. Plos Pathogens (2022). Also, the three references concerning N protein (6,7,8) are not specific to the protein N, these references are just very general works about coronavirus and SARS-CoV-2.
-The figure 2 is misleading due to different scales for the y-axis , it seems that the -Log10 pvalue are larger for human than for bats, the y-scale must be the same
-in the text line 322 it is mentioned that DDX 21 is a DEAD box helicase, from the table S4 it is DExD box helicase
Reviewer 2 Report
Comments and Suggestions for Authors
This work my Armstrong et al. investigates the binding of the SARS-CoV-2 Nucleocapsid (N) to host proteins when expressed in human (293T) and fruit bat (RoNI/7) cells. Using a combination of IP-Mass Spec and bioinformatics, the authors identify numerous common and unique protein-protein interactions with SARS2 N and host proteins in the two lines. In addition, bioinformatic tools are used to identify common host pathways among the identified proteins. Together, these data help elucidate differences in SARS2 N protein function between bats and humans.
Overall, the research appears sound and is mostly ok for publication as is. While limited in scope, the amount of work and quality of the data is mostly in line with the expectations of the journal. The findings themselves are moderately informative, and while follow up experiments are necessary to determine to what extent differences in SARS2 N protein-protein interactions affect infection, I do not believe such studies are necessary for a short article in Viruses
The only requested change is that I am uncomfortable approving an article where ChatGPT was used to provide gene descriptions (see Table S3). While generative AI may be the wave of the future, and to the authors' credit they labeled the descriptions clearly, the accuracy of LLM is currently suspect. These descriptions should be removed.
